# WILT: A Multi-turn, Memorization-Robust Inductive Logic Benchmark for LLMs

**Eryk Banatt**
Riot Games
Los Angeles, CA 90064
ebanatt@riotgames.com

**Jonathan Cheng**
Riot Games
Los Angeles, CA 90064
joncheng@riotgames.com

**Tiffany Hwu**
Riot Games
Los Angeles, CA 90064
thwu@riotgames.com

## Abstract

While large language models (LLMs) have shown impressive capabilities across a wide range of domains, they still encounter significant challenges in reasoning tasks that require gathering evidence over multiple turns and drawing logical conclusions from this evidence. Despite the multi-turn nature of many real-world LLM use cases, most existing benchmarks rely on carefully curated single-turn tests, which often blur the line between memorization and genuine reasoning. To address this, we introduce the **Wason Inductive Logic Test (WILT)**, a simple yet challenging multi-turn reasoning benchmark designed to resist memorization. WILT is inspired by the Wason 2-4-6 task [41], where participants must infer a basic boolean function involving three variables (e.g., $x < y < z$) by proposing test cases (such as $(2, 4, 6)$). In WILT, each test starts from a clean slate, with only the initial instructions provided, preventing models from relying on pre-learned responses. Our findings reveal that LLMs struggle with this task, with the best-performing model achieving only 28% accuracy, highlighting a significant gap in LLM performance on complex multi-turn reasoning tasks.

## 1 Introduction

Large Language Models (LLMs) powered by the transformer architecture [36] have enabled a new computing paradigm powered by natural language. LLMs have seen an increased ubiquitousness in day-to-day life beyond the machine learning research space, where they help many people across the world with common tasks. These models interact with users through *multi-turn conversations*, a capability added to next-token-prediction models via instruction-tuning [19] and alignment post-training phases [25].

Measuring the performance of LLMs has been challenging for the research community. Publicly available benchmarks are subject to Goodhart's law [12] or implicit overfitting [27], and difficult benchmarks often keep a held-out, publicly unavailable test set to accurately evaluate models [5]. In addition, the vast majority of benchmarks are suites of single-turn tests, which differ substantially from the common day-to-day use of LLMs [15] [16].

The reasoning capability of LLMs, particularly in multi-turn scenarios, is of substantial interest. A commonly reported failure pattern for LLMs is the "doom loop", where the model repeatedly responds with a near-identical message to one of its earlier messages, providing minimal utility. While some benchmarks have emerged to attempt to measure this multi-turn propensity of repetition [18], none so far have done so for multi-step inductive reasoning.

In this work, we make the following contributions:

38th Conference on Neural Information Processing Systems (NeurIPS 2024).

1. We introduce the **Wason Inductive Logic Test (WILT)**, a multi-turn inductive logic benchmark that is robust to memorization. We show that frontier LLMs struggle significantly on this task.

2. We further evaluate the reasoning capabilities of the LLMs by framing the task as an exploration of hypothesis space. The LLMs show marked differences in the rate of hypothesis space reduction, novelty of responses, and complexity.

## 2 WILT

The Wason Inductive Logic Test (WILT) is a benchmark for LLM reasoning inspired by the Wason 2-4-6 task [41]. Models begin with the instruction that they must uncover the hidden rule, and may pose up to 30 test cases of that rule. For example, they can pose the tuple $(2, 4, 6)$ and the test will respond with "True, 29 Attempts Remaining."

All hidden rules take three numbers and return a boolean. These rules are simple and non-stochastic, so there is no additional value to posing the same test multiple times. Valid inputs include any float or integer that can be typed in three or fewer characters, excluding signs and the decimal point (e.g. -999, 1.23, 5). The hidden rules are written as Python lambda functions. After a maximum of thirty tries (or any turn before then), the model may make one attempt to guess the function, after which the test will terminate. The model must return a Python lambda function that is the same as or equivalent[1] to the hidden rule in order to receive full points.

WILT is conceptually simple, but very challenging. Humans can identify simple rules fairly easily despite the infinitely large hypothesis space, the unbounded difficulty of a hidden function, and the impossibility of verifying the correctness of your response. For example, consider the canonical Wason task rule of $x < y < z$. This rule has very high overlap with the much more arbitrary rule $(x < y < z) \land (x \neq 12)$. The WILT benchmark therefore tests a few high-value behaviors of interest:

1. **Multi-Turn Capability**: Participants that fall into "doom loops" are punished by virtue of having less useful information with which to infer the hidden rule.

2. **Hypothesis Space Reduction**: Participants are rewarded for proposing test cases that effectively narrow down the possible rules, despite that hypothesis space being infinitely large.

3. **Susceptibility to Confirmation Bias**: Participants who are more prone to "confirming their hypothesis" rather than seeking falsification will perform poorly upon this task.

4. **Inductive Mathematical Reasoning**: Proposing good test cases is a useful test of inductive reasoning and the ability to generalize from a number of specific examples.

5. **Deductive Mathematical Reasoning**: Proposing sensible functions after observing many test cases is a useful test of deductive reasoning, with successful performance rewarding identifying a specific function that fits a suite of examples.

6. **Occam's Razor**: Participants are rewarded for finding the simplest explanation fitting the examples.

We release two test suites: a *lite split*, with 10 very easy tests and a canonical *full split* with 50 moderately difficult tests. Future work will extend this to include a procedurally generated split for robustness to overfitting. We find that the lite split quickly produces a roughly similar ordering to the full split, but we report results upon the full split for the remainder of this work. Please see the appendix for further details. The test and associated code can be found at github.com/riotgames/wilt.

## 3 Related Work

Compared to other reasoning benchmarks, WILT stands out as both highly multi-turn focused and unusually robust to memorization. In contrast to other benchmarks, WILT requires models to *interact with an environment* by proposing their own test cases to uncover a hidden function without relying on pre-provided examples. This setup reduces the risk of overfitting, as each test begins with the same initial instructions, and the model must generate and interpret its own data.

---

[1]For example, $x - y = z$ is equivalent to $y + z = x$

## 3.1 Reasoning Benchmarks

There are a wide variety of reasoning benchmarks used to evaluate large language models. Some very notable among these are MATH [16], GSM8K [6], CommonsenseQA [30], StrategyQA [11], BIG-BENCH [29], SciBench [39], SVAMP [26], ARC-AGI [5], MMLU [15], GPQA [28], and HumanEval [4]. These benchmarks are the standard for measuring LLM reasoning capabilities, but are overwhelmingly carefully chosen single-turn problems which aim to meaningfully separate the performance of different models on reasoning-like outputs such as math, code, or logic puzzles. However, these benchmarks are subject to train-on-test leakage, even if efforts are made to decontaminate the dataset [45], and the majority are explicitly single-turn tests. Our benchmark directly measures the model's ability to navigate multi-turn scenarios and does not require careful hiding of a test set to prevent misleading results.

With respect to reasoning about simple functions, a benchmark that stands out as similar to ours is CRUXEval [14], which assembles a list of 800 simple Python functions and input-output pairs, evaluating language models on their ability to predict input from output and output from input. Our work could be seen as a multi-turn, more difficult extension of this work – one where the function is replaced with a black box, where helpful and informative input-output pairs are not provided but instead need to be searched for by the language model, and where the objective is to infer the hidden function rather than the input or output.

## 3.2 Multi-Turn Benchmarks

There are a handful of multi-turn benchmarks used to evaluate LLMs. PlanBench [34] is one prominent benchmark that attempts to measure the ability of LLMs to navigate planning problems. This is a class of problems that is solved easily by classical planning algorithms such as STRIPS [10], and like our benchmark poses a significant challenge to LLMs. PlanBench is a primarily multi-step, single-turn benchmark with a multi-turn component (i.e. replanning based on unexpected events), which contrasts with our benchmark's more direct multi-turn focus. This can be observed in the o1 models performing comparatively well on PlanBench [35], since scaling inference time compute within a single turn would be expected to improve performance substantially. Similarly, BIG-BENCH [29] includes a "20 Questions" task, which pairs two language models together to guess a word by posing questions about it. Like ours, this involves an iterated reduction of hypothesis space over multiple turns, but unlike ours relies on potentially subjective or unreliable LLM responses to queries about the concept.

Closest to ours are MINT [40] and Aidan-bench [18], which have more direct multi-turn focus. MINT repurposes existing single-turn benchmarks by allowing models to use tools before answering. While the range of tasks in MINT is therefore quite large, strong models can still solve these tasks in few (or one) turns, and the unmodified prompts remain subject to test set leakage. Aidan-bench measures the cosine similarity between multi-turn responses. This represents a more pure measurement of the doom loop phenomenon. In our benchmark, rather than directly measuring the doom loops, we are instead measuring how often those doom loops lead to failures of reasoning. We see similar performances in our benchmark compared to Aidan-bench (e.g. Mistral Large), but with an ordering more tied to capabilities (e.g. Sonnet's strong results, see Table 1).

## 3.3 Hypothesis Space Reduction

Hypothesis space representation is a commonly used framing in inductive logic tasks for LLMs. In [38], the authors show a technique called *hypothesis search* where the model will propose hypotheses in natural language and then implement these hypotheses as Python programs. This technique was shown to improve performance on ARC-AGI [5], but a similar approach could be used along with chain-of-thought [42] for WILT as well.

## 4 Results

Our results for this test can be found in Table 1. We show that despite the test's relative simplicity, most models struggle substantially both to propose good tests and to infer a rule based on available evidence. As in the original Wason 2-4-6 task, we find a common failure mode to be confirmation

Table 1: Model Accuracy Comparison

| Model | Accuracy | Approx. Correct | Avg. Guesses | Repeats |
|---|---|---|---|---|
| Claude 3.5 Sonnet 20240620 [2] | **14/50** | **10/50** | 16.38 | 27 |
| o1-mini 2024-09-12 [24] | 13/50 | 8/50 | 12.1 | **3** |
| o1-preview 2024-09-12 [24] | 12/50 | 6/50 | **8.12**[2] | **3** |
| chatgpt-4o-latest [22] | 11/50 | 7/50 | 14.22 | 38 |
| Mistral Large 2 [20] | 11/50 | 5/50 | 26.56 | 142 |
| GPT-4o 2024-08-06 [22] | 9/50 | 6/50 | 15.26 | 26 |
| Llama 3.1 405B [8] | 8/50 | 9/50 | 12.21 | 30 |
| Gemini 1.5 Flash 0827 [13] | 7/50 | 4/50 | 14.04 | 108 |
| Llama 3.1 70B [8] | 7/50 | 2/50 | 15.18 | 74 |
| Deepseek-v2.5-chat [17] | 6/50 | 5/50 | 27.22 | 489 |
| GPT-4o-mini [23] | 6/50 | 2/50 | 20.36 | 54 |
| Gemini 1.5 Pro [13] | 5/50 | 6/50 | 16.78 | 41 |
| Gemini 1.5 Flash [13] | 5/50 | 6/50 | 16.5 | 123 |
| Deepseek-v2-coder [46] | 5/50 | 5/50 | 21.82 | 335 |
| Deepseek-v2-chat [17] | 3/50 | 3/50 | 25.32 | 334 |
| Llama 3.1 8b [8] | 3/50 | 0/50 | 26.18 | 223 |
| Open Mistral Nemo [21] | 2/50 | 3/50 | 27.34 | 400 |
| Claude 3 Haiku 20240307 [3] | 1/50 | 1/50 | 6.76 | 11 |
| Gemini 1.5 Flash 8b 0827 [13] | 0/50 | 2/50 | 26.76 | 386 |
| Gemma 2 9B [31] | 0/50 | 2/50 | 8.82 | 70 |

bias – a participant will identify a plausible hypothesis and continue to propose tests that attempt to *confirm* it. Stronger models will more explicitly attempt to *falsify* these hypotheses instead.

In Table 1, we include a column *approximately correct*, measuring the number of rules in which the model was able to correctly identify some critical behavior of the rule, but returned a rule with failing edge cases. For example, guessing $(x < y < z)$ instead of $(x \leq y \leq z)$ is approximately correct. We include this column to highlight models that are more willing to guess immediately instead of uncovering edge cases by default (e.g. Llama 3.1 405B). In these cases, we could see potentially improved performance through more targeted prompting techniques.

We find that LLMs (particularly smaller ones) will frequently repeat tests they have already used, sometimes dozens of times. We therefore also provide a column *repeats* which counts the total proposed tests which were already tested for that rule. Further discussion on test novelty can be found in Appendix A.4. Additionally, further experiments analyzing what parts of the task models are strong or weak at can be found in Appendix A.2. We find that some models are produce very useful test cases (e.g. chatgpt-4o-latest) whereas other models are strong at guessing the rule (e.g. o1-mini).

## 4.1 Hypothesis Space Reduction

To compare the LLMs' ability to efficiently reason about the task, we estimate how effectively each model reduces the hypothesis space [38]. At best, an LLM should always propose a triplet that eliminates as many untested hypotheses as possible. At worst, a model repeatedly proposes a triplet confirming previously covered hypotheses. For example, an LLM that has already guessed $(2, 4, 6)$ retreads much of the same hypothesis space by guessing $(4, 6, 8)$ rather than $(0.01, -1, 100)$.

To represent that hypothesis space, we randomly generate a large number of lambdas that encompass a wide range of potential hypotheses. For example, we randomly generate lambdas having to do with: ordering $(x < y < z)$, equality $(x = y \neq z)$, arithmetic relations $(x + y = z)$, parity $(x \leq 10, y \leq 5, z \leq 100)$, etc. When an LLM proposes a triplet, we cross off any lambdas that do not match the observed behavior of the hidden rule. Figure 1 illustrates the rate at which different models reduce the hypothesis space over successive turns. Models with worse reasoning spend more attempts to clear less of the hypothesis space.

---

[2]We bold results for this column only if the models are also high performing. For example, Claude 3 Haiku uses fewer guesses than o1-preview, but this is because it is failing.

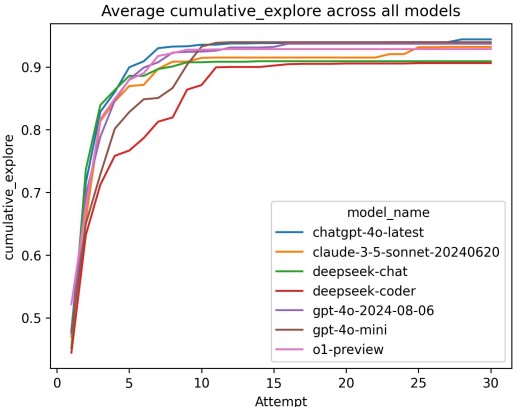

Figure 1: Models can succeed upon this task by reducing the hypothesis space quickly or providing useful tests for many turns. We show that models with strong reasoning capabilities can narrow the space quickly, but weaker multi-turn capability harms their ability to get value out of later tests.

## 4.2 Response Complexity

To determine how well the models employ Occam's Razor, we explore different metrics to gauge whether the models find the simplest rule that covers the examples. From existing Bayesian models of cognition [32, 33], the *size principle* uses hypothesis size as a measure of simplicity. In these Bayesian models, hypothesis size is calculated as the number of values that match the hypothesis. Calculating hypothesis size in this manner is only possible when the test values are within a limited countable range. In our case, the possible test values are infinite, requiring some alternative metrics to gauge hypothesis size. We use three metrics:

1. **Number of Operators**: We count the number of operators used in the rule expression.
2. **Response Length**: We calculate the string length of the rule expression. The longer the length, the more complex it is likely to be. As longer outputs tend to be arbitrarily preferred by automatic evaluators [9], it is particularly important to measure the brevity of the response for cases in which simplicity is desired.
3. **Set Inclusion**: We generate a grid of integer-float tuples and apply them to guessed and actual rules to generate sets of tuples returning "True". If the set of the guessed rule is a subset or superset of the actual rule, we then calculate their set size ratio. A ratio of $1$ is ideal, $> 1$ suggests a less complex guess, and $< 1$ a more complex one.

Appendix A.3 shows the complexity metrics of the LLMs. Most LLMs with high accuracy have long response lengths and many operators, with some exceptions.

## 5 Discussion

In our experiments, we show that LLMs struggle substantially with this task. Specifically, their propensity to repeat test cases, propose useless test cases, and guess very unlikely rules harms their performance on this task substantially. The varying performance in a multi-turn setting represents a previously underappreciated dimension of measuring reasoning capability in LLMs. There is much work in language modeling for code-based agents [7] [43] and LLM-driven unit testing [44], and the difficulty of LLMs to explore edge cases effectively has substantial implications on those applications.

With this work, we aim to provide a benchmark that measures a model's capacity for *exploring an environment* and reasoning based on its own decisions across multiple turns. We believe that this paradigm offers a more direct measurement of reasoning capability compared to other benchmarks. By focusing specifically on how LLMs handle multi-turn reasoning, we better understand their most common real-world applications.

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

# A    Appendix / Supplemental Material

## A.1    Experimental Details

### A.1.1    Model Configurations

Models are accessed when possible with default configurations and settings provided by their respective proprietary APIs (i.e. Anthropic, OpenAI, Mistral, DeepSeek, and Vertex APIs). Llama 3.1 models are run via the Amazon Bedrock API. Gemma 2 9b and Llama 3 70B are run via the Groq API [1].

### A.1.2    Prompts

All models begin from the same initial prompt.

> You are an AI tasked with discovering a hidden rule that takes three numbers as input and returns a boolean. These numbers can be any float or integer number that can be typed in 3 or fewer digits, not including signs and decimal characters (e.g. -11.3 is a valid value). In each response, you will provide your step-by-step reasoning for your future plans on this task, and then you will take one of two actions:
>
> 1. Provide a single test case to observe the output of a rule (e.g. "Test Case: ```(1, 2, 3)```" which corresponds to x=1, y=2, z=3. You may only propose one test case at a time.
> 2. Provide a final "guess" in the form of a Python lambda function (e.g. "Final Guess: ```lambda x,y,z: x < y < z```"). You only get to select this action one time, so don't use it unless you are confident or are forced to.
>
> You will have up to 30 attempts at test cases, and you will have exactly one attempt at guessing the final function. Make sure all of the tests you provide are unique from other ones you have already tried – the functions are all deterministic, and will be the same every time. You should prioritize getting the rule as correct as possible, since an incorrect answer will get no points. You should therefore not guess the function unless you are quite confident, or if you are very close to running out of attempts.
>
> You must return your responses in the format laid out above at the very bottom of your message. For example, if you want to submit a test case, you must conclude with the string "Test Case: ```(x,y,z)```", where x,y,z are replaced with your guesses. If you want to submit a final guess, you must conclude with the string "Final Guess: ```<function>```" where <function> is replaced with a Python lambda function. Do not include any comments or additional text on the same lines as these two things.
>
> Make sure to include your reasoning for your tests – what you are testing for, why you selected that test, etc.

Responses by the models are pulled out via regular expressions matching the formatting in the prompt. We find that after controlling for various formatting eccentricities (Python blocks, markdown, bold characters, etc) that all listed models are capable of providing test cases in this format.

### A.1.3    Verifying Equivalent Functions

To verify two provided lambda functions are equivalent, we generate a large number of test cases and ensure the provided rules match on all outputs. Specifically, we create three sets of cases:

1. **Integer Grid Cases** - We construct a 40x40x40 grid of integer triplets from -20 to 20, inclusive, leading to 64,000 triplet cases.
2. **Random Uniform Cases** - We construct a list of 10,000 uniformly random float triplets from -200 to 200, inclusive.

3. **Special Cases** - We hand-design a small set of test cases to ensure all hidden rules in the full split are adequately tested.

We mark a rule as incorrect if any test cases generated above show different behavior between the hidden rule and the guessed rule, and mark it correct otherwise.

## A.2 Evaluating Function Inversion Capability

To succeed at the WILT task, models must succeed at both gathering evidence (hypothesis reduction) and drawing logical conclusions from evidence (function inversion). To distinguish a model's ability to do one or the other, we perform an experiment where models attempt to guess the rule using tests from another model. Rather than asking the model to gather evidence, we directly provide it all the reasoning-stripped[3] input-output pairs generated by another model for the same rule, and ask the model to guess the rule in a single turn. Without the original reasoning and subsequent observation before and after each test case, we expect most models to underperform relative to the full test even when provided their own cases. Likewise, we expect models stronger at single-turn to perform better in this experiment relative to other models subject to the same evidence. Our results can be found in Figure 2.

This reveals some notably varied capabilities among the top performing models. While Claude Sonnet 3.5 was the narrowly highest performing model on the full test, this experiment reveals important context for why that may be. We see that it performs better than most other models subject to the same evidence, but proposes test cases that are generally slightly less useful for other models without the attached justifications. Likewise, without its own reasoning for each case, Sonnet's performance degraded substantially more than other models in the same setting, suggesting a larger component of its success was its reasoning, compared to the test cases alone.

o1-mini shows highly superior single-turn capability in this test, but notably performs relatively less well when provided its own tests rather than the tests of another high-performing model. When paired with cases from chatgpt-4o-latest, it successfully guessed 19 of the 50 rules, far surpassing the best-performing single model in the full test.

Despite having many repeated tests and messages which were generally similar to each other (see Tables 1 and 3), we see that Mistral Large performs well with other models' tests and provides a corpus of tests useful to other models. We note its comparable performance to chatgpt-4o-latest both along the rows (model's performance with other model's test cases) and columns (performance of other models with the model's test cases), reinforcing its strong performance in the full test.

Critically, we show that models have non-identical strengths and weaknesses, and that success on the full WILT task depends on strong performance on a few key metrics of interest. Even without the attached reasoning for test cases, composing the test case generation of one model and the function inversion of another model very often outperforms using a single strong model for both subtasks. This has some notable implications for future LLM applications: in [37] it was shown that several language models coordinated by an aggregator LLM could outperform strong single models. Future work could explore coordinating models for both single-turn and multi-turn oriented tasks, potentially leading to improved performance.

---

[3]We strip reasoning to avoid conflating confirmation bias in the attached reasoning, rather than just the accumulated evidence.

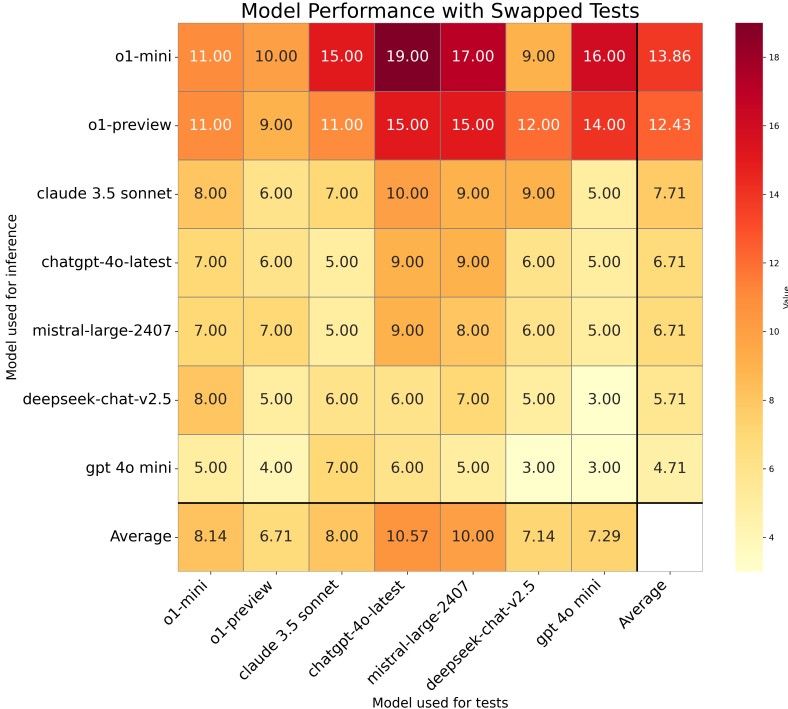

Figure 2: Models have varying success when using test cases proposed by other models. o1-mini stands out as having much stronger single-turn reasoning in this experiment, but it performs poorly with its own tests.

## A.3 Response Complexity Evaluation

Table 2: Response Complexity (Median)

| Model | Num Operators | Response Length | Set Inclusion |
|-------|--------------|-----------------|---------------|
| Claude 3.5 Sonnet | 3 | 34.5 | 0.08 |
| o1-mini 2024-09-12 | 3 | 29.0 | 0.79 |
| o1-preview-2024-09-12 | 2 | 25 | 0.01 |
| chatgpt-4o-latest | 5 | 39 | 1.0 |
| Mistral Large 2 | 5 | 39 | 1.0 |
| GPT-4o 2024-08-06 | 4.5 | 39 | 0.34 |
| Llama 3.1 405B | 2 | 30 | 0.52 |
| Gemini 1.5 Flash 0827 | 4 | 35.5 | 0.00 |
| Llama 3.1 70B | 2 | 25 | 1.00 |
| Deepseek-v2.5-chat | 3 | 29 | 0.27 |
| GPT-4o-mini | 5 | 39.5 | 0.05 |
| Gemini 1.5 Pro | 3 | 38 | 0.27 |
| Gemini 1.5 Flash | 3 | 28 | 0.06 |
| Deepseek-v2-coder | 5 | 39 | 0.88 |
| Deepseek-v2-chat | 2 | 28 | 0.00 |
| Llama 3.1 8b | 2 | 23 | 0.05 |
| Open Mistral Nemo | 5 | 46 | 1.00 |
| Claude 3 Haiku | 5 | 31 | 0.49 |
| Gemini 1.5 Flash 8b 0827 | 3 | 29 | 0.02 |
| Gemma 2 9B | 5 | 38 | 0.52 |

Table 2 shows the complexity of each model by the three aforementioned metrics.

In Figure 3 we show the set inclusion ratios in the case where a model is provided another model's test cases. That is, we show whether an error in the final guess of a model is likely to be smaller / less than one (e.g. $x < y < z$ instead of $x \leq y \leq z$), or larger / greater than one (e.g. $x > 0$ instead of $x > 0 \wedge x < 5$). This seems more test-case dependent than other complexity benchmarks, where tests provided by certain models seem to lead to smaller hypotheses.

Figure 4 shows the length of the string used to guess the rule by each model, which is comparatively more consistent for a model. We find this to be fairly consistent across settings, with most models hovering near 45.

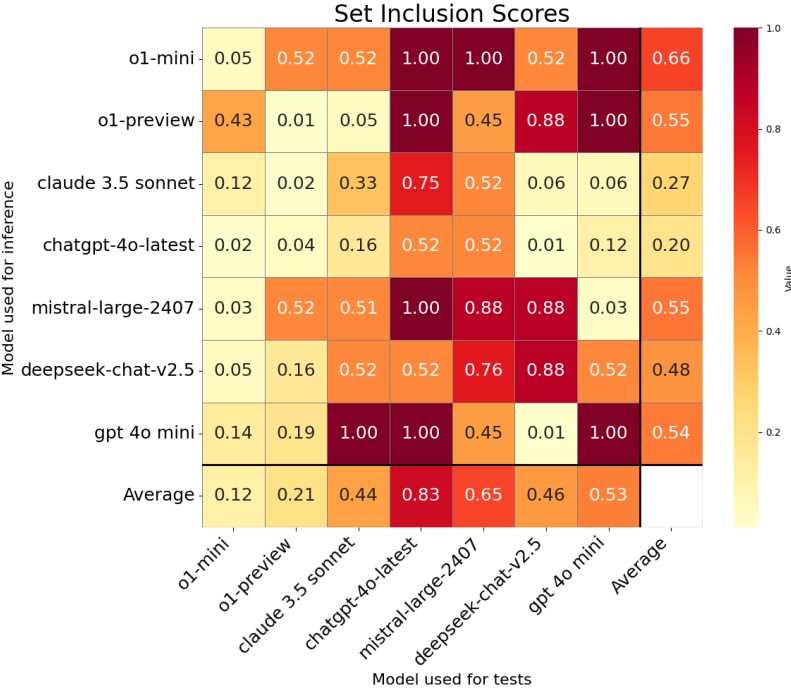

Figure 3: Set inclusion ratios will differ in the same model when provided another model's tests, and models provided the same tests have different set inclusion behaviors.

Figure 5 shows the median number of operators used by a model when given another model's test cases, as in Section A.2. This can be used to estimate the model's bias toward guessing a simpler rule. o1-preview, for example, tends to use fewer operators than the other models subject to the same evidence. This also highlights potential discrepancies in the complexity resulting from a model's test cases and the complexity of its final guesses. When models use test cases generated by DeepSeek Chat v2.5, they tend to use fewer operators, likely because the test cases are fully encompassed by simple rules like `lambda x,y,z: False`. Conversely, when given other models' tests, DeepSeek Chat v2.5 responds with a high level of complexity compared to other models. Its guesses often overfit a complicated rule to the test cases (e.g., it guesses a rule of `lambda x, y, z: y == x or y == z or y == (x + z) / 2` when given the following true test cases for even numbers: (2.0, 2.0, 2.0), (2.0, 2.0, 4.0), (−2.0, 4.0, 6.0), (0.0, 2.0, 4.0), (2.2, 4.0, 6.0), (2.0, 4.0, 6.0).)

### A.4 Test Case Novelty

Test case novelty is an interesting second order metric for success upon the WILT task. Broadly speaking, models that reuse fewer tests are rewarded with more information for which to solve the task. Models that very rarely re-propose a test tend to perform very well upon WILT, but the inverse is not necessarily true – models that loop tests often still arrive at the right answer.

Repeated tests are useful for bifurcating the types of failures on WILT – one being the doom loop phenomenon, and the other being reasoning capability conditioned upon some available evidence.

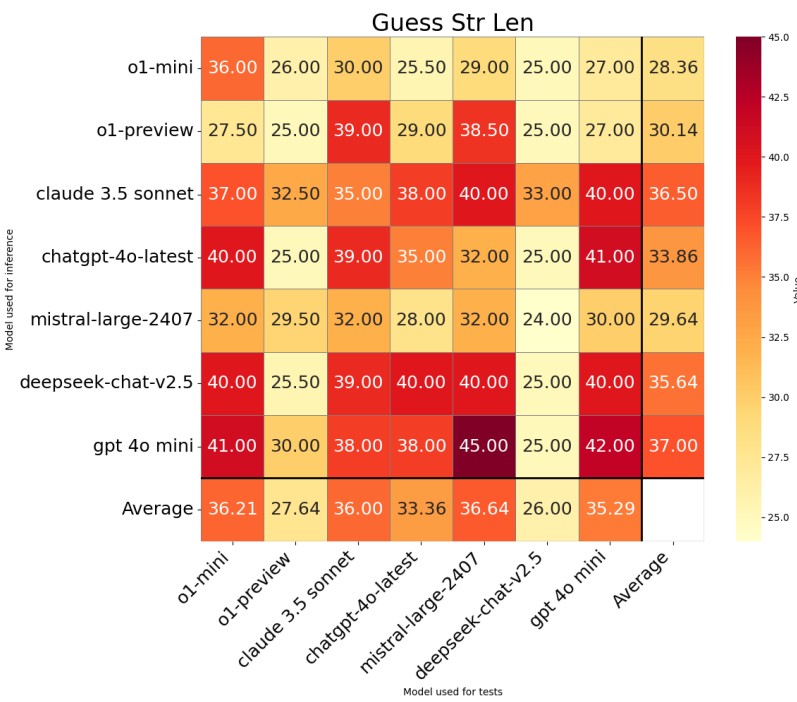

Figure 4: Models tend to have fairly consistent guess string lengths, with some exceptions.

One hypothesis for the observed behavior is that certain models are primarily oriented towards single-turn scenarios, and that one type of failure need not imply the other. DeepSeek Chat v2.5, for example, demonstrates strong initial hypothesis space reduction compared to other models, which allows it notably better performance on WILT compared to other models with similar repeat counts (e.g. open-mistral-nemo). Strong single-turn performance and deductive reasoning capabilities can help salvage performance from a model that demonstrates difficulty with multi-turn inductive logic.

Following Aidan-bench [18], we provide Table 3, containing additional novelty metrics. These include:

1. *Average novelty* - which reports the average cosine similarity between each message's gpt-3 embeddings and the closest previous message within the same test.

2. *Average minimum novelty* - which reports the average *minimum* cosine similarity between each message's gpt-3 embeddings and the closest previous message within the same test.

These capture an additional dimension of "test novelty" and "message novelty", where models may propose the same tests for different reasons, or repeat previously generated messages verbatim. We bold results which are best within the class of high performing models. We note that models that propose fewer tests before guessing (e.g. o1-preview, o1-mini) should see lower values for all of these compared to models that tend to use many tests before guessing (e.g. mistral-large) even for otherwise equally performing models. We also show the average novelty by turn in Figure 6, which captures the model's novelty scores across turns.

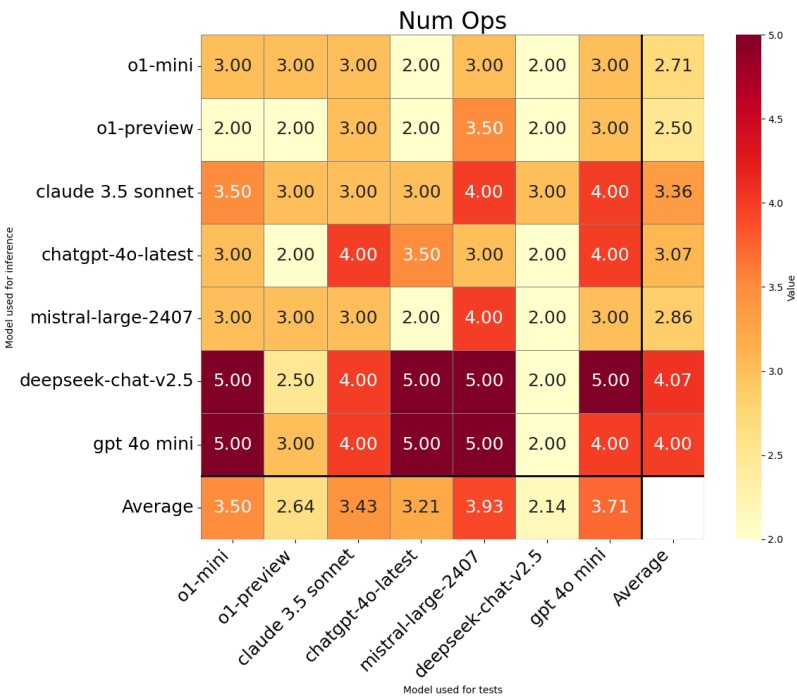

Figure 5: The median number of operators provides a window into guessed rules being more or less complicated compared to guesses provided by other models.

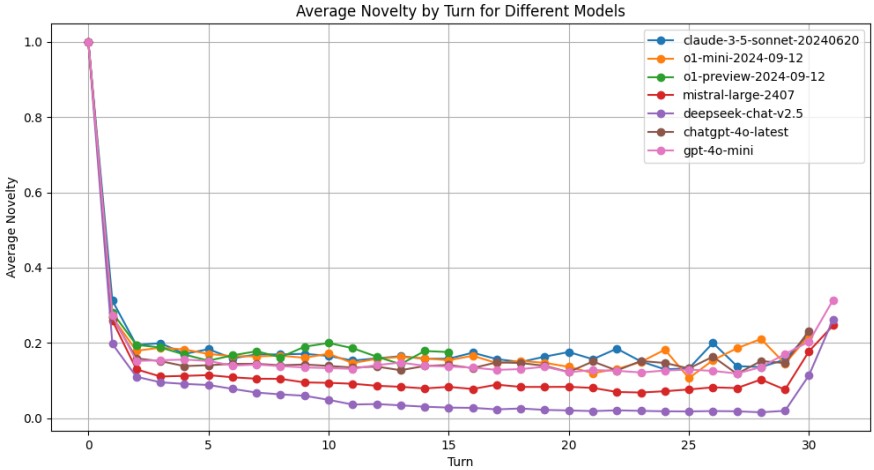

Figure 6: Cosine similarity by turn for selected models. Models have a higher novelty score near the end, since the final guess is often much different from previous messages, which are all proposed test cases.

## A.5  Selected Lite Split Results

Table 4 contains selected results for the *lite split*. We show this much easier split produces a similar ordering, suggesting that the bulk of the separation in our benchmark lies in the easier tests.

Table 3: Model Novelty Metrics

| Model | Repeats | Avg. Novelty | Avg. Min. Novelty |
|---|---|---|---|
| claude-3.5-sonnet-20240620 | 27 | 0.22 | 0.08 |
| o1-mini-2024-09-12 | **3** | 0.24 | 0.09 |
| o1-preview-2024-09-12 | **3** | **0.28** | **0.11** |
| chatgpt-4o-latest | 38 | 0.21 | 0.10 |
| mistral-large-2407 | 142 | 0.13 | 0.04 |
| gpt-4o-2024-08-06 | 26 | 0.22 | 0.11 |
| llama3.1 405B | 30 | 0.20 | 0.06 |
| gemini-1.5-flash-exp-0827 | 108 | 0.19 | 0.08 |
| llama3-70b | 74 | 0.19 | 0.06 |
| deepseek-chat-v2.5 | 489 | 0.09 | 0.02 |
| gpt-4o-mini | 54 | 0.19 | 0.10 |
| gemini-1.5-pro | 41 | 0.26 | 0.13 |
| gemini-1.5-flash | 123 | 0.19 | 0.08 |
| deepseek-coder | 334 | 0.11 | 0.03 |
| deepseek-chat-v2 | 335 | 0.10 | 0.03 |
| llama-3.1-8b | 223 | 0.13 | 0.01 |
| open-mistral-nemo | 400 | 0.12 | 0.02 |
| claude-3-haiku-20240307 | 11 | 0.29 | 0.11 |
| gemini-1.5-flash-8b-exp-0827 | 386 | 0.14 | 0.05 |
| gemma2-9b-it | 70 | 0.34 | 0.18 |

Table 4: Lite Split Metrics

| Model | Accuracy | Avg. Guesses |
|---|---|---|
| Claude-3.5-Sonnet | 8/10 | 13.68 |
| GPT-4-Turbo | 7/10 | 12.47 |
| GPT-4o | 6/10 | 13.86 |
| DeepSeek-V2-Coder | 6/10 | 23.46 |
| Llama 3 70B | 4/10 | 15.64 |
| DeepSeek-V2-Chat | 2/10 | 24.82 |
| Llama 3 8B | 1/10 | 24.0 |
| GPT-3.5-Turbo | 1/10 | 2.9 |

## A.6 Test Descriptions

Table 5 contains the functions found in the *lite split* of WILT. Table 6 contains the functions found in the *full split* of WILT. These are implemented as Python lambda functions.

Table 5: Lite Split: Complete Set of Tests

| Test ID | Description |
|---|---|
| 1 | $x > y > z$ |
| 2 | $x < y < z$ |
| 3 | $x \geq y \geq z$ |
| 4 | $x \leq y \leq z$ |
| 5 | $x = y = z$ |
| 6 | $x \neq y \wedge y \neq z \wedge x \neq z$ |
| 7 | $x < 0 \wedge y < 0 \wedge z < 0$ |
| 8 | $x + y = z$ |
| 9 | $x \cdot y = z$ |
| 10 | $x < y \wedge y > z$ |

# B  Failure Case Examples

## B.1  Doom Loop Without Reasoning Failure

Table 7 shows an example of a model (Deepseek-Chat-v2.5) entering a doom loop during the test proposal phase, but where that doom loop does not constitute a reasoning failure. Reasoning has been removed for brevity.

## B.2  Approximately Correct

Table 8 shows an example of o1-mini getting a very difficult test case (co-primality) approximately correct, failing only because it adds an additional arbitrary constraint upon the magnitude of the values despite no such constraint existing. Reasoning has been removed for brevity.

## B.3  Confirmation Bias

Table 9 shows an example of o1-preview failing a relatively easy test case ($x \geq y \geq z$) due to a confirmation bias error. The model uses only 9 test cases and correctly identifies that the rule returns true when all three are equal, but submits five test cases confirming that and none exploring other rules which are true when three items are equal. Reasoning has been removed for brevity.

## B.4  Same Test For New Reason

Table 10 shows an example of Claude Sonnet 3.5 repeating a test, where it will mistakenly generate the same test for a different stated reason. We see the model notice it has repeated a test only after it has already submitted the test. Other tests have been removed for brevity.

Table 6: Full Split: Complete Set of Tests

| Test ID | Description |
|---|---|
| | **Easy Tests** |
| 1 | $x > y > z$ |
| 2 | $x < y < z$ |
| 3 | $x \geq y \geq z$ |
| 4 | $x \leq y \leq z$ |
| 5 | $x < z < y$ |
| 6 | $x \leq z \leq y$ |
| 7 | $z < x < y$ |
| 8 | $z \leq x \leq y$ |
| 9 | $x = y = z$ |
| 10 | $x \neq y \wedge y \neq z \wedge x \neq z$ |
| 11 | $x < 0 \wedge y < 0 \wedge z < 0$ |
| 12 | $x > 0 \wedge y > 0 \wedge z > 0$ |
| 13 | $x \bmod 2 = 0 \wedge y \bmod 2 = 0 \wedge z \bmod 2 = 0$ |
| 14 | $x \bmod 2 \neq 0 \wedge y \bmod 2 \neq 0 \wedge z \bmod 2 \neq 0$ |
| | **Medium Tests** |
| 15 | $x + y = z$ |
| 16 | $x \cdot y = z$ |
| 17 | $x + z = y$ |
| 18 | $x \cdot z = y$ |
| 19 | $y + z = x$ |
| 20 | $y \cdot z = x$ |
| 21 | $\max(x, y, z) = x$ |
| 22 | $\max(x, y, z) = y$ |
| 23 | $\max(x, y, z) = z$ |
| 24 | $\min(x, y, z) = x$ |
| 25 | $\min(x, y, z) = y$ |
| 26 | $\min(x, y, z) = z$ |
| 27 | $x + y + z = 0$ |
| 28 | $x \cdot y \cdot z = 0$ |
| 29 | $(x + y + z) \bmod 2 = 0$ |
| 30 | $(x + y + z) \bmod 2 = 1$ |
| 31 | $(x \cdot y \cdot z) \bmod 2 = 0$ |
| 32 | $(x \cdot y \cdot z) \bmod 2 = 1$ |
| 33 | $(x + y)/2 = z$ |
| 34 | $-5 \leq x, y, z \leq 5$ |
| 35 | $-10 \leq x, y, z \leq 10$ |
| 36 | $-5 \leq x, y, z \leq 0$ |
| 37 | $0 \leq x, y, z \leq 5$ |
| 38 | $-2 \leq x, y, z \leq 2$ |
| 39 | $-20 \leq x, y, z \leq 20$ |
| | **Very Hard Tests** |
| 40 | $x^2 + y^2 = z^2$ |
| 41 | $x^2 + z^2 = y^2$ |
| 42 | $y^2 + z^2 = x^2$ |
| 43 | $\lfloor x \rfloor \& \lfloor y \rfloor = \lfloor z \rfloor$ (bitwise AND) |
| 44 | $\lfloor x \rfloor | \lfloor y \rfloor = \lfloor z \rfloor$ (bitwise OR) |
| 45 | $\lfloor x \rfloor \oplus \lfloor y \rfloor = \lfloor z \rfloor$ (bitwise XOR) |
| 46 | $x, y, z \in \mathbb{Z} \wedge \gcd(x, y) = \gcd(y, z) = \gcd(z, x) = 1$ |
| 47 | $\sqrt{\lfloor |x| \rfloor}^2 = \lfloor |x| \rfloor \wedge \sqrt{\lfloor |y| \rfloor}^2 = \lfloor |y| \rfloor \wedge \sqrt{\lfloor |z| \rfloor}^2 = \lfloor |z| \rfloor$ |
| 48 | $0 < x \bmod 1 \wedge 0 < y \bmod 1 \wedge 0 < z \bmod 1$ |
| 49 | $0 < x \bmod 1 < y \bmod 1 < z \bmod 1 < 1$ |
| 50 | $x < y < z \wedge 0 < z - x \leq 1$ |

Table 7: DeepSeek-v2.5 Enters a doom loop but still arrives at the correct answer

| Turn | Assistant | Harness |
|---|---|---|
| 1 | Test Case: (1, 1, 1) | (1.0, 1.0, 1.0): True. |
| 2 | Test Case: (1, 2, 3) | (1.0, 2.0, 3.0): True. |
| 3 | Test Case: (3, 2, 1) | (3.0, 2.0, 1.0): True. |
| 4 | Test Case: (2, 1, 3) | (2.0, 1.0, 3.0): True. |
| 5 | Test Case: (-1, -2, -3) | (-1.0, -2.0, -3.0): False. |
| 6 | Test Case: (-1, 2, -3) | (-1.0, 2.0, -3.0): False. |
| 7 | Test Case: (0, 1, 2) | (0.0, 1.0, 2.0): False. |
| 8 | Test Case: (2, 3, 1) | (2.0, 3.0, 1.0): True. |
| 9 | Test Case: (0, 1, 1) | (0.0, 1.0, 1.0): False. |
| 10 | Test Case: (0.001, 1, 1) | (0.001, 1.0, 1.0): True. |
| 11 | Test Case: (-0.001, 1, 1) | (-0.001, 1.0, 1.0): False. |
| 12 | Test Case: (999, 1, 1) | (999.0, 1.0, 1.0): True. |
| 13 | Test Case: (-999, 1, 1) | (-999.0, 1.0, 1.0): False. |
| 14 | Test Case: (0.0001, 1, 1) | (0.0001, 1.0, 1.0): True. |
| 15 | Test Case: (-0.0001, 1, 1) | (-0.0001, 1.0, 1.0): False. |
| 16 | Test Case: (999.999, 1, 1) | (999.999, 1.0, 1.0): True. |
| 17 | Test Case: (-999.999, 1, 1) | (-999.999, 1.0, 1.0): False. |
| 18 | Test Case: (0.0001, 999.999, 1) | (0.0001, 999.999, 1.0): True. |
| 19 | Test Case: (-0.0001, 999.999, 1) | (-0.0001, 999.999, 1.0): False. |
| 20 | Test Case: (0.0001, -999.999, 1) | (0.0001, -999.999, 1.0): False. |
| 21 | Test Case: (0.0001, 999.999, -0.0001) | (0.0001, 999.999, -0.0001): False. |
| 22 | Test Case: (0.0001, 999.999, -999.999) | (0.0001, 999.999, -999.999): False. |
| 23 | Test Case: (0.0001, 999.999, 0.0001) | (0.0001, 999.999, 0.0001): True. |
| 24 | Test Case: (0.0001, 0.0001, 999.999) | (0.0001, 0.0001, 999.999): True. |
| 25 | Test Case: (999.999, 0.0001, 0.0001) | (999.999, 0.0001, 0.0001): True. |
| 26 | Test Case: (0.0001, 999.999, 0.0001) | (0.0001, 999.999, 0.0001): True. |
| 27 | Test Case: (0.0001, 0.0001, 999.999) | (0.0001, 0.0001, 999.999): True. |
| 28 | Test Case: (999.999, 0.0001, 0.0001) | (999.999, 0.0001, 0.0001): True. |
| 29 | Test Case: (0.0001, 999.999, 0.0001) | (0.0001, 999.999, 0.0001): True. |
| 30 | Test Case: (0.0001, 0.0001, 999.999) | (0.0001, 0.0001, 999.999): True. |
| 31 | Final Guess: lambda x, y, z: $x > 0$ and $y > 0$ and $z > 0$ | Congratulations! Your guess is correct. |

Table 8: o1-mini nearly gets the coprimality case correct, but adds bounds for no concrete reason.

| Turn | Assistant | Harness |
|---|---|---|
| 1 | Test Case: (1, 2, 3) | (1.0, 2.0, 3.0): True. |
| 2 | Test Case: (3, 2, 1) | (3.0, 2.0, 1.0): True. |
| 3 | Test Case: (1, 1, 1) | (1.0, 1.0, 1.0): True. |
| 4 | Test Case: (-1, 2, 3) | (-1.0, 2.0, 3.0): True. |
| 5 | Test Case: (-1, -2, -3) | (-1.0, -2.0, -3.0): True. |
| 6 | Test Case: (0, 0, 0) | (0.0, 0.0, 0.0): False. |
| 7 | Test Case: (0, 1, 2) | (0.0, 1.0, 2.0): False. |
| 8 | Test Case: (1.5, 2.5, 3.5) | (1.5, 2.5, 3.5): False. |
| 9 | Test Case: (1, 2, 3.0) | (1.0, 2.0, 3.0): True. |
| 10 | Test Case: (1, 2, 3.1) | (1.0, 2.0, 3.1): False. |
| 11 | Test Case: (1, 2, -3) | (1.0, 2.0, -3.0): True. |
| 12 | Test Case: (10, -20, 30) | (10.0, -20.0, 30.0): False. |
| 13 | Test Case: (4, 5, 6) | (4.0, 5.0, 6.0): False. |
| 14 | Test Case: (-2, -1, 3) | (-2.0, -1.0, 3.0): True. |
| 15 | Test Case: (3, 3, 4) | (3.0, 3.0, 4.0): False. |
| 16 | Test Case: (1, -1, 2) | (1.0, -1.0, 2.0): True. |
| 17 | Test Case: (-3, 3, -2) | (-3.0, 3.0, -2.0): False. |
| 18 | Test Case: (2, -2, 2) | (2.0, -2.0, 2.0): False. |
| 19 | Test Case: (-1, 2, -2) | (-1.0, 2.0, -2.0): False. |
| 20 | Test Case: (1, 3, 2) | (1.0, 3.0, 2.0): True. |
| 21 | Test Case: (1, 2, 2) | (1.0, 2.0, 2.0): False. |
| 22 | Test Case: (-1, -2, 3) | (-1.0, -2.0, 3.0): True. |
| 23 | Test Case: (2, 3, -1) | (2.0, 3.0, -1.0): True. |
| 24 | Final Guess: lambda x, y, z: all(isinstance(n, int) and $n \neq 0$ and $-3 \leq n \leq 3$ for n in [x, y, z]) and math.gcd(x, y) == 1 and math.gcd(x, z) == 1 and math.gcd(y, z) == 1 | Sorry, that's not the correct rule. |

Table 9: o1-preview fails early on an easy test case despite having a wealth of remaining tests, because it latched on to an early behavior and sought to confirm it rather than falsify it.

| Turn | Assistant | Harness |
|------|-----------|---------|
| 1 | Test Case: (2, 3, 6) | (2.0, 3.0, 6.0): False. |
| 2 | Test Case: (2, 3, 5) | (2.0, 3.0, 5.0): False. |
| 3 | Test Case: (1, 1, 1) | (1.0, 1.0, 1.0): True. |
| 4 | Test Case: (2, 2, 2) | (2.0, 2.0, 2.0): True. |
| 5 | Test Case: (1, 1, 2) | (1.0, 1.0, 2.0): False. |
| 6 | Test Case: (-1, -1, -1) | (-1.0, -1.0, -1.0): True. |
| 7 | Test Case: (1.5, 1.5, 1.5) | (1.5, 1.5, 1.5): True. |
| 8 | Test Case: (0, 0, 0) | (0.0, 0.0, 0.0): True. |
| 9 | Test Case: (1, 2, 3) | (1.0, 2.0, 3.0): False. |
| 10 | Final Guess: lambda x, y, z: x == y == z | Sorry, that's not the correct rule. |

Table 10: Claude Sonnet 3.5 double submits $(2, 3, 5)$ to test $x + y = z$ and $all(isprime(n))$

| Turn | Assistant | Harness |
|------|-----------|---------|
| 1-5 | ... | ... |
| 6 | Still False. Let's try a different approach and consider some mathematical relationships. Reasoning for the next test: 1. Let's test if the rule involves a simple arithmetic relationship. 2. We'll use numbers where the third is the sum of the first two. 3. We'll use small integers for clarity. Test Case: (2, 3, 5) | (2.0, 3.0, 5.0): False. |
| 7-8 | ... | ... |
| 9 | We're still not getting a True result. Let's try a different approach and consider some number theory concepts. Reasoning for the next test: 1. Let's test if the rule involves prime numbers. 2. We'll use three consecutive prime numbers. 3. This will also test if there's a special relationship with consecutive numbers that are not simply ascending. Test Case: (2, 3, 5) | (2.0, 3.0, 5.0): False. |
| 10 | We've already tested (2, 3, 5) before, and it's still False. Let's try a different number theory concept... | ... |

# C Full Conversation Examples

## C.1 Doom Loop on Coding Task

Below we provide a full example of a "doom loop" interacting with an LLM in a multi-turn conversation in a real world coding task. We use Llama 3 8B to generate code for a pytorch variational autoencoder, and after 2 turns of debugging it repeats an already mentioned suggestion.

### User:

Can you write me some code which trains a VAE on cifar10 using pytorch?

### Llama3-8b-8192:

Here's a basic example of how you can train a Variational Autoencoder (VAE) on CIFAR-10 using PyTorch:

```Python
import torch
import torch.nn as nn
import torch.optim as optim
from torchvision import datasets, transforms
from torch.utils.data import DataLoader
import numpy as np

# Define the VAE model
class VAE(nn.Module):
    def __init__(self):
        super(VAE, self).__init__()
        self.encoder = nn.Sequential(
            nn.Conv2d(3, 32, kernel_size=3),
            nn.ReLU(),
            nn.MaxPool2d(2, 2),
            nn.Conv2d(32, 64, kernel_size=3),
            nn.ReLU(),
            nn.MaxPool2d(2, 2),
            nn.Flatten()
        )
        self.fc1 = nn.Linear(64*4*4, 128)
        self.fc2 = nn.Linear(128, 128)
        self.fc3 = nn.Linear(128, 64*4*4)
        self.fc4 = nn.Linear(64*4*4, 3*32*32)

    def reparameterize(self, mu, logvar):
        std = torch.exp(0.5*logvar)
        eps = torch.randn_like(std)
        return mu + eps*std

    def encode(self, x):
        h1 = self.encoder(x)
        h2 = torch.relu(self.fc1(h1))
        h3 = torch.relu(self.fc2(h2))
        mu = self.fc3(h3)
        logvar = self.fc4(h3)
        return mu, logvar

    def decode(self, z):
        h1 = torch.relu(self.fc4(z))
        h2 = torch.relu(self.fc3(h1))
        h3 = torch.relu(self.fc2(h2))
        h4 = torch.relu(self.fc1(h3))
```

```
            return torch.sigmoid(self.fc4(h4))

    def forward(self, x):
        mu, logvar = self.encode(x)
        z = self.reparameterize(mu, logvar)
        return self.decode(z), mu, logvar

# Define the loss function
def loss_function(recon_x, x, mu, logvar):
    BCE = nn.functional.binary_cross_entropy(recon_x, x.view(-1,
    ↪   3*32*32), reduction='sum')
    KLD = 0.5 * torch.sum(1 + logvar - mu.pow(2) - logvar.exp())
    return BCE + KLD

# Load the CIFAR-10 dataset
transform = transforms.Compose([transforms.ToTensor()])
trainset = datasets.CIFAR10(root='./data', train=True, download=True,
↪   transform=transform)
trainloader = DataLoader(trainset, batch_size=64, shuffle=True)

testset = datasets.CIFAR10(root='./data', train=False, download=True,
↪   transform=transform)
testloader = DataLoader(testset, batch_size=64, shuffle=False)

# Initialize the VAE model and the optimizer
vae = VAE()
optimizer = optim.Adam(vae.parameters(), lr=0.001)

# Train the VAE
for epoch in range(100):
    for i, (x, _) in enumerate(trainloader):
        x = x.to(device)
        optimizer.zero_grad()
        recon_x, mu, logvar = vae(x)
        loss = loss_function(recon_x, x, mu, logvar)
        loss.backward()
        optimizer.step()
        if i % 100 == 0:
            print('Epoch: %d, Batch: %d, Loss: %.3f' % (epoch+1, i+1,
            ↪   loss.item()))

    # Test the VAE
    test_loss = 0
    with torch.no_grad():
        for x, _ in testloader:
            x = x.to(device)
            recon_x, mu, logvar = vae(x)
            loss = loss_function(recon_x, x, mu, logvar)
            test_loss += loss.item()
    test_loss /= len(testloader)
    print('Test Loss: %.3f' % test_loss)
```

This code defines a VAE model, loads the CIFAR-10 dataset, and trains the model using the Adam optimizer. The loss function is the sum of the binary cross-entropy loss and the Kullback-Leibler divergence. The model is trained for 100 epochs, and the test loss is printed after each epoch.

Note that you need to install the 'torchvision' package to use the CIFAR-10 dataset. You can install it using 'pip install torchvision'. Also, you need to specify the device (GPU or CPU) where you want to run the model. You can do this by adding

```
`device = torch.device('cuda' if torch.cuda.is_available()
else 'cpu')` before the training loop.
```

### User:

I'm getting the following error trying to run this code:

```
RuntimeError Traceback (most recent call last)
<ipython-input-3-471d3f454439> in <cell line: 70>()
     72          x = x.to(device)
     73          optimizer.zero_grad()
---> 74          recon_x, mu, logvar = vae(x)
     75          loss = loss_function(recon_x, x,
                                      mu, logvar)
     76          loss.backward()

6 frames
in forward(self, input)
    115
    116     def forward(self, input: Tensor) -> Tensor:
--> 117         return F.linear(input, self.weight, self.bias)
    118
    119     def extra_repr(self) -> str:

RuntimeError: mat1 and mat2 shapes cannot be multiplied (64x2304 and
↪   1024x128)
```

### llama3-8b-8192:

The error message indicates that there is a mismatch between the shapes of the input and the weights in one of the fully connected layers in your VAE model.

The error message specifically says that the shapes are 64x2304 and 1024x128, which means that the input has a shape of (64, 2304) and the weights have a shape of (1024, 128).

The issue is likely due to the fact that the input to the fully connected layer is not what you expect. In the 'encode' method of your VAE model, you have the following lines:

```
h1 = self.encoder(x)
h2 = torch.relu(self.fc1(h1))
h3 = torch.relu(self.fc2(h2))
mu = self.fc3(h3)
logvar = self.fc4(h3)
```

The 'h3' tensor has a shape of (64, 128) because 'self.fc2' is a fully connected layer with 128 output units. However, 'self.fc3' and 'self.fc4' are also fully connected layers, and they are expecting an input with a shape of (batch_size, 128) where batch_size is the number of samples in the batch.

In your case, the batch size is 64, so the input to 'self.fc3' and 'self.fc4' should be (64, 128), not (64, 128) as it is now.

To fix this issue, you can try to flatten the output of 'self.encoder' before passing it to the fully connected layers. You can do this by adding the following line after 'h1 = self.encoder(x)':

```
h1 = h1.view(-1, 64*4*4)
```

This will flatten the output of 'self.encoder' into a tensor with a shape of (batch_size, 64*4*4), which should be compatible with the weights of

'self.fc1', 'self.fc2', 'self.fc3', and 'self.fc4'.

### User:

Okay, I added that line. Now I'm getting a similar error with different values:

```
RuntimeError: mat1 and mat2 shapes cannot be multiplied
(144x128 and 1024x3072)
```

### llama3-8b-8192:

The error message indicates that there is a mismatch between the shapes of the input and the weights in one of the fully connected layers in your VAE model.

The error message specifically says that the shapes are 144x128 and 1024x3072, which means that the input has a shape of (144, 128) and the weights have a shape of (1024, 3072).

The issue is likely due to the fact that the input to the fully connected layer is not what you expect. In the 'encode' method of your VAE model, you have the following lines:

```
h1 = self.encoder(x)
h2 = torch.relu(self.fc1(h1))
h3 = torch.relu(self.fc2(h2))
mu = self.fc3(h3)
logvar = self.fc4(h3)
```

The 'h3' tensor has a shape of (64, 128) because 'self.fc2' is a fully connected layer with 128 output units. However, 'self.fc3' and 'self.fc4' are also fully connected layers, and they are expecting an input with a shape of (batch_size, 128) where batch_size is the number of samples in the batch.

In your case, the batch size is 64, so the input to 'self.fc3' and 'self.fc4' should be (64, 128), not (64, 128) as it is now.

To fix this issue, you can try to flatten the output of 'self.encoder' before passing it to the fully connected layers. You can do this by adding the following line after 'h1 = self.encoder(x)':

```
h1 = h1.view(-1, 128)
```

This will flatten the output of 'self.encoder' into a tensor with a shape of (batch_size, 128), which should be compatible with the weights of 'self.fc1', 'self.fc2', 'self.fc3', and 'self.fc4'.

However, the error message also mentions that the input has a shape of 144x128, which is not compatible with the weights of the fully connected layer. This suggests that there may be an issue with the architecture of your VAE model.

To fix this issue, you can try to adjust the architecture of your VAE model to ensure that the input to the fully connected layers is compatible with the weights.

