# OpenReview forum: "WILT: A Multi-turn, Memorization-Robust Inductive Logic Benchmark for LLMs"
_NeurIPS.cc/2024/Workshop/MATH-AI — MATH-AI 24_

### Official Review · Reviewer_YxGM · 2024-09-28

**Rating:** 7
**Confidence:** 3

**Review:**

**Summary:** The authors propose a new benchmark dataset: WILT. Each task in this dataset is a Boolean algebra expression and the model needs to infer what this expression is by querying some oracle. Models’ responses are then rated through a host of complexity metrics: set inclusion, response length, number of operations. The authors demonstrate that WILT is difficult for existing LLMs and claim that it is inherently susceptible for memorization or overfitting.

**Quality:** 3 good.

**Clarity:** 3 good. From the writing itself, this reviewer can understand the main ideas of the paper, the experimental setup as well as the motivation of the paper.

**Originality:** 3 good. Though there are existing interactive benchmarks for reasoning (such as twenty questions from BIG bench), this task is relatively novel.

**Significance:** 3 good. The idea proposed by the authors is a general one. This reviewer can easily see this setup being generalized to more complex tasks and settings.

**Pros:**
- The behaviors of interest that the authors identify are cogent. They capture many fundamental processes of inductive reasoning.
- Though the size of the dataset is small, this reviewer can easily imagine how this can be fixed by simply crafting more lambda expressions.
- The authors benchmark their dataset against a wide range of the newest models providing solid experimental evidence that this benchmark is useful.

**Cons:**
- Though this reviewer understands the page length is limited, a review of the related literature is lacking. Specifically, there are little comparisons with existing interactive reasoning benchmarks. In this reviewer’s opinion, the setup here is analogous to a hybrid of tool use benchmarks (where the LLM understand how to craft useful queries to the oracle) [1] and interactive reasoning benchmarks (understand how to update its world view given new information) [2]. It would be nice to discuss this more.

[1] https://arxiv.org/pdf/2310.03128

[2] https://github.com/google/BIG-bench/tree/main/bigbench/benchmark_tasks/twenty_questions

---

### Official Review · Reviewer_F8bb · 2024-10-01

**Rating:** 9
**Confidence:** 4

**Review:**

This paper proposes a new benchmark called the Wason inductive logic test (WILT) for testing the multi-turn reasoning ability of large language models.  Intuitively, this can be seen as an interactive inductive logic programming task that a language model can feed a test case to the environment and get back a response saying yes or no, at the final stage the model will guess the hidden rule of this logic test.

This task is very interesting because it tests the ability of the language model to do induction. The result of this benchmark is also significant, we can see that almost none of the current LLM can do this task well.

Overall, I think this is a nice work and strongly recommend to accept it, but I still have one question:

About *Occam's Razor*: It is quite natural that there are multiple rules that can satisfy the same observations in a limited number of tries, so in this case, what if the LLM returns a plausible explanation that satisfies all the test cases they have quried, but still not 100% same with the underlying ground truth? How can we measure which rule is simpler?

---

### Official Review · Reviewer_1Q4a · 2024-10-05
**a significant step forward in the design of benchmarks for evaluating the reasoning capabilities of LLMs in multi-turn scenarios.**

**Rating:** 7
**Confidence:** 4

**Review:**

The paper introduces the Wason Inductive Logic Test (WILT), a new benchmark for large language models (LLMs) focused on multi-turn, memorization-resistant inductive logic. This benchmark, inspired by the classical Wason 2-4-6 task, challenges LLMs to infer a hidden boolean function through up to thirty test cases. The models begin with no initial clues and must propose tests to discover the underlying rule, encoded as a Python lambda function. This setup aims to evaluate and enhance LLMs' capabilities in multi-turn reasoning and hypothesis testing, areas where current models often struggle.

Strengths:

1.	WILT is designed to be robust against memorization, requiring models to actively participate in a reasoning process rather than recalling fixed responses, addressing a critical limitation in current LLM evaluations.

2.	The benchmark assesses multiple aspects of cognitive reasoning, including the application of Occam's Razor, avoidance of repetitive loops, and efficient navigation of large hypothesis spaces.

3.	By focusing on multi-turn interactions, WILT mirrors realistic use-cases of LLMs, making its findings applicable to practical settings where LLMs must understand and manipulate complex, evolving scenarios.

Weaknesses:

1.	The benchmark's reliance on synthetic, logical constructs might not adequately represent the variety of real-world logic and reasoning challenges, potentially biasing the evaluation towards tasks that are well-suited to current AI capabilities rather than pushing the boundaries of what LLMs can achieve.

2.	The use of approximations and subjective judgments in determining whether a model's response is "approximately correct" introduces ambiguity into the assessment process, which could affect the benchmark's reliability and the consistency of its outcomes across different studies.

---

### Official Review · Reviewer_nQ6u · 2024-10-08

**Rating:** 6
**Confidence:** 4

**Review:**

#### Summary:
The paper introduces WILT, a multi-turn reasoning benchmark inspired by the Wason 2-4-6 task, designed to test inductive reasoning capabilities of LLMs. The benchmark focuses on multi-turn interactions where models must infer a hidden boolean function by proposing test cases. WILT aims to be robust against memorization and emphasizes inductive and deductive reasoning, hypothesis space reduction, and susceptibility to confirmation bias. The paper demonstrates that existing LLMs struggle with this task and provides empirical results comparing the performance of several models, highlighting significant differences in reasoning abilities across them.

#### Strengths:
1. **Interesting Problem**: The paper tackles a novel problem of multi-turn inductive reasoning that hasn't been extensively studied in previous benchmarks. By modeling the benchmark after the classic Wason 2-4-6 task, it introduces a challenging reasoning scenario that is particularly difficult for LLMs, pushing the boundaries of reasoning in AI systems.

2. **Sufficient Number of Models for Evaluation**: The authors evaluate a wide range of models, from Claude and GPT-4 variants to Mistral and Gemini models, providing a comprehensive analysis of their performance across different reasoning metrics such as hypothesis space reduction and response complexity.

#### Weaknesses:
1. **Lack of Insights**: While the paper does a good job of presenting the empirical results, it falls short in providing deeper insights or theoretical analyses of why models struggle with WILT. A more detailed discussion on the nature of the failures and what specific aspects of model architectures or training regimes contribute to their performance deficits would significantly strengthen the paper's contributions.

2. **Absence of Proven Prompting Methods**: The paper doesn't incorporate or experiment with prompting techniques such as chain-of-thought (CoT) reasoning, which has been shown in prior works to improve reasoning tasks in LLMs. Including these methods in the evaluations would have provided a more thorough assessment of LLM reasoning capabilities and potentially enhanced model performance on WILT.

---

### Decision · Program_Chairs · 2024-10-09

Accept